

# Plume-exit modeling to determine cloud condensation nuclei activity of aerosols from residential biofuel combustion

**Francisco Mena[1], Tami C. Bond[1], Nicole Riemer[2]**

[1]Department of Civil and Environmental Engineering, University of Illinois Urbana-Champaign, Urbana, IL 61801, USA
[2]Department of Atmospheric Sciences, University of Illinois Urbana-Champaign, Urbana, IL 61801, USA

*Correspondence to*: Tami Bond (yark@illinois.edu)

**Abstract.** Residential biofuel combustion is an important source of aerosols and gases in the atmosphere. The change in cloud characteristics due to biofuel burning aerosols is uncertain, in part, due to the uncertainty in the added number of cloud condensation nuclei (CCN) from biofuel burning. We provide estimates of the CCN activity of biofuel burning aerosols by
explicitly modeling plume dynamics (coagulation, condensation, chemical reactions, and dilution) in a young biofuel burning plume from emission until plume-exit, defined here as the condition when the plume reaches ambient temperature and specific humidity through entrainment. We found that aerosol-scale dynamics affect CCN activity only during the first few seconds of evolution, after which the CCN efficiency reaches a constant value. Homogenizing factors in a plume are co-emission of SVOCs or emission at small particle sizes; SVOC co-emission can be the main factor determining plume-exit CCN for
hydrophobic or small particles. Coagulation limits emission of CCN to about $10^{16}$ per kg of fuel. Depending on emission factor, particle size, and composition, some of these particles may not activate at low $s_{sat}$. Hygroscopic Aitken mode particles can contribute to CCN through self-coagulation, but have a small effect on the CCN activity of accumulation-mode particles, regardless of composition differences. Simple models (monodisperse coagulation and average hygroscopicity) can be used to estimate plume-exit CCN within about 20% if particles are unimodal and have homogeneous composition, or when particles
are emitted in the Aitken mode even if they are not homogeneous. On the other hand, if externally-mixed particles are emitted in the accumulation mode without SVOCs, an average hygroscopicity overestimates emitted CCN by up to a factor of 2. This work has identified conditions under which particle populations become more homogeneous during plume processes. This homogenizing effect requires the components to be truly co-emitted, rather than sequentially emitted.

## 1.   Introduction

Residential combustion of biofuels (including wood, dung, agricultural residues, and charcoal) is an important source of trace gases that play a role in atmospheric chemistry (Chuen-Yu et al., 2011; Williams et al., 2012). Ludwig et al. (2003) estimated that domestic biofuel combustion contributes 7-20% of the global budget of $CO_2$, CO, and NO. Akagi et al. (2011) estimated that biofuel burning emits 150 Tg/yr of non-methane organic compounds.



Biofuel combustion is also an important source of particles, contributing 20% of primary organic carbon (OC) and black carbon (BC) globally (Bond et al., 2004). These particles affect radiative transfer through interactions with sunlight and clouds. Scattering and absorption of solar radiation by aerosol particles is known as the "direct effect" of aerosols on climate, while the "indirect effect" refers to changes in radiation as aerosol particles alter cloud characteristics. The indirect effect can counteract the warming due to greenhouse gases and absorbing aerosols by increasing the cloud albedo.

Improving model estimates of Earth's radiative response to changes in human activities requires a better understanding of how aerosols with different compositions and mixing states can act as CCN and, hence, influence cloud properties. The overall climate impact of biofuel combustion aerosols is still uncertain because, although biofuel emissions contain BC, an important warming agent (Jacobson, 2000), the magnitude of the indirect effect of biofuel combustion aerosols is still unknown. Jacobson (2010) found that eliminating biofuel soot and gases in addition to fossil fuel soot emissions resulted in a cooling effect and reduced surface air temperatures by $0.4 - 0.7$ K, whereas eliminating fossil fuel soot emissions alone decreased air temperatures by only $0.3 - 0.5$ K. In contrast, Bauer et al. (2010) found that reducing biofuel emissions of black and organic carbon aerosols by 50% decreased the number concentration of cloud condensation nuclei (CCN), leading to an increase in radiative forcing of 0.13 W/m$^2$, and therefore led to an overall warming effect. Koch et al. (2011) summarized multiple model results showing that eliminating biofuel burning soot would reduce CCN concentrations and lead to a warming effect. The reason for the discrepancy in these studies lies, in part, in estimating the indirect effect. Both Bauer et al. (2010) and Koch et al. (2011) highlighted the sensitivity of their results to assumptions about aerosol properties such as size distribution and mixing state, that is, how chemical species are distributed across the particle population. Mann et al. (2014) compared models with observations under the AeroCom initiative and reported that some differences might be attributable to missing particle growth or assumptions in size distribution of emissions. Seinfeld et al. (2016) identified the necessity of assessing the level of detail in aerosol microphysical and chemical properties required to represent aerosol-cloud interactions.

In this work, we examine combustion of biofuel to provide energy rather than open biomass burning. The latter refers to large open fires of natural origin or for deforestation and agricultural clearing. Although both activities combust cellulosic fuels, residential biofuel burning plumes dilute to ambient conditions within seconds to minutes. In contrast, biomass burning plumes from an individual event result from uncontrolled combustion of a large area of fuel, and can extend for several kilometers (Trentmann et al., 2006). Open biomass burning contributes 74% of primary OC and 42% of BC to the global budget (Bond et al., 2004). Biofuel burning contributes a smaller fraction (20% of primary OC and BC globally) but represents still a large fraction of global emissions that dominates some regions.

In global models of atmospheric chemistry, emissions are instantaneously diluted into the large grid boxes in which the emissions occur. Therefore, plume-scale processes of coagulation, condensation and evaporation, chemical reactions, entrainment of dry air, and dilution, are neglected (Poppe et al., 1998; Gao et al., 2003; Jost et al., 2003), although they may affect the properties of aerosols in the plume (Trentmann et al., 2005). In young plumes, defined here as conditions where





within-plume concentrations and temperatures are greatly elevated above background, high gas and particle concentrations intensify these processes. Within the first hour, it is common to observe increases in inorganic aerosol mass by factors of 2-10 (Gao et al., 2003; Hobbs et al., 2003), and increases in organic aerosol mass by factors of 1.6-4 (Alvarado and Prinn, 2009; Vakkari et al., 2014). Chemical and physical processes that occur during the early evolution of the plume affect aerosol composition, size, and hygroscopicity, and, thereby, the aerosols' ability to become cloud droplets, known as activation. Research on activation properties within young plumes has examined large-scale biomass burning plumes (*e.g.*, Mason et al. 2001; Trentmann et al. 2002, 2005, 2006; Gao et al. 2003; Jost et al. 2003; Alvarado et al., 2009). To our knowledge, no study has investigated the plume dynamics that affect aerosol evolution within small-scale, young biofuel burning plumes.

The objective of this study is to address this research gap for the specific case of residential biofuel combustion. We model the evolution of particles within biofuel burning plumes from emission until plume exit, defined here as the condition when the plume reaches ambient temperature and specific humidity. These simulations are performed for a range of combustion and plume scenarios to estimate likely plume-exit properties and to understand the level of detail required for estimating CCN concentrations.

## 2. Methodology

A young plume is a rapidly changing environment. After emission, the plume rises due to thermal convection, and entrainment of background air lowers the plume temperature as well as the gas and particle concentration. The decrease in temperature increases relative humidity (RH), but the entrainment of background air counteracts this by decreasing the water content (Trentmann et al., 2006; Shrivastava et al., 2006). The following sections explain how we modeled each of these processes.

### 2.1. Modeling aerosol evolution and CCN activation: PartMC-MOSAIC

We modeled the aerosol dynamics within the plume with the Particle Monte Carlo (PartMC) model (Riemer et al., 2009) coupled to the Model for Simulating Aerosol Interactions and Chemistry (MOSAIC) (Zaveri et al., 2008). PartMC-MOSAIC is a box model that tracks the composition of individual particles within a Lagrangian air parcel as they evolve due to Brownian coagulation, condensation and evaporation (*i.e.* gas-particle partitioning), chemical reactions, and dilution. Whereas coagulation and dilution are modeled as stochastic processes with a Poisson distribution, MOSAIC simulates the gas and particle-phase thermodynamics, and gas-particle mass transfer (Riemer et al., 2009; Zaveri et al., 2008). MOSAIC uses the SORGAM scheme to model the formation of low- and semi-volatility products from the oxidation of precursors, and their gas-particle partitioning (Schell et al., 2001; Zaveri et al., 2010). PartMC-MOSAIC simulates the mixing ratio of 77 gases and the mass concentration of 20 aerosol species in the plume. For each simulation we used a time-step of 0.005 s and $10^5$ computational particles, each one representing a number of simulated particles in the plume.

The high concentrations of gases and particles in a biofuel burning plume, as well as rapid temperature changes, lead to rapid condensation of semi-volatile species and aerosol coagulation. Since PartMC-MOSAIC is a particle-based method, which fully



resolves the composition of a representative sample of individual particles in the plume, it is suited to study the evolution of complex biofuel burning aerosol distributions, and to investigate changes that may not be captured by moment-based (Whitby et al., 1997, McGraw et al., 1997) or sectional (Jacobson, 1997) schemes. While the latter two schemes have a long tradition in aerosol modeling studies, they approximate particle size and composition, and hence, their predictions are limited by the

accuracy of the approximation. For scenarios where mixing state does not have an effect on particle properties, these approaches are sufficient; for instance, if mixing by coagulation can be neglected relative to the condensation rate and no fresh emissions are entering the aerosol population, then all particles evolve equally and a sectional or moment-based scheme will capture the particle evolution. However, in the rapid dilution of young biofuel burning plumes, high coagulation and condensation rates affect particles' composition and size. Capturing the evolution of the mixing state may be important for

predicting CCN activity. To obtain confidence in modeled outcomes in a rapidly-changing environment, it is necessary to use a scheme, such as a particle-resolved model, that avoids a priori approximations for the particle mixing state and size, hence PartMC-MOSAIC is a uniquely-suited tool for our study.

PartMC-MOSAIC has been used to model aerosol dynamics in urban areas (Riemer et al., 2009, 2010; Zaveri et al., 2010), CCN activation (Ching et al., 2012; Fierce et al., 2013) and ship plumes (Tian et al., 2014). Within PartMC-MOSAIC, we

also implemented a module that reproduces the entrainment of background air (Jones, 2014), which allows for a variable water content in the plume.

PartMC-MOSAIC uses prescribed temporal profiles of plume dilution rate, specific humidity, and temperature. Their derivation is detailed in Sect. 2.2. The initial conditions for the simulations are the particle size distribution, the particle composition at emission, and the gas mixing ratios in the plume at emission. To examine plume behavior and sensitivities,

these initial conditions are specified and varied as described in Sect. 2.3. In the following section we explain how PartMC-MOSAIC outputs can be used to describe the evolution of the CCN activity of biofuel burning particles.

### 2.1.1. CCN activation

PartMC-MOSAIC returns the composition of individual particles in the plume for each time step of the simulation, which we use to calculate the hygroscopicity of each particle. We use the $\kappa$-Köhler model here to calculate particle hygroscopicity

(Petters and Kreidenweis, 2007). The hygroscopicity parameter $\kappa$ comes from a parameterization of Köhler theory (Petters and Kreidenweis, 2007) and represents the hygroscopicity of a substance, that is, its capacity to uptake water. The value of $\kappa$ for a particle is calculated as the volume-average of the $\kappa$ of each aerosol component. With the value of $\kappa$ and the particle size, the critical supersaturation at which each particle activates can be calculated using the $\kappa$-Köhler model

We also use several metrics to quantify CCN activity. First, the CCN efficiency is the fraction of particles that activate at a

given supersaturation ($s_{sat}$). At a sufficiently large $s_{sat}$, all the particles activate and the CCN efficiency is one. Secondly, we calculate the critical supersaturation of the particle population, consisting of the supersaturation at which 5% ($ssat_5$), 50%





(ssat$_{50}$), and 95% (ssat$_{95}$) of the plume particles activate. We also evaluate the Emission Index of CCN (EI$_{CCN}$), which is the total number of CCN present at plume-exit per mass of fuel burnt (units of kg$^{-1}$).

In laboratory or field measurements the particle population is not sampled directly at the flame but after a period of cooling, so we use 325 K as a reference temperature. We label parameters at this temperature with a '325' subscript, *e.g.* the particle number concentration N$_{325}$, or count median diameter CMD$_{325}$. We also calculate the number of CCN at plume-exit normalized to the number of particles at a plume temperature of 325 K, which we term CCN$_{exit}$/N$_{325}$. This parameter indicates the fraction of emitted particles that result in cloud-active particles, accounting for losses due to coagulation and lack of activity due to small size or hydrophobic nature. Similarly, we calculate the fraction of plume particles that reach plume-exit (N$_{exit}$/N$_{325}$). In the calculation, we normalize initial and final plume concentrations by the concentration of carbon monoxide (CO) to account for the plume dilution. Since coagulation is modeled as a stochastic process, repeated simulations are not identical, but EI$_{CCN}$ differences are within about 3%.

### 2.2. Prescribed plume profiles used in PartMC-MOSAIC

#### 2.2.1. Plume entrainment

Buoyancy-induced convection induces both vertical and horizontal dispersion (Trentmann et al., 2005, 2006). A higher dilution rate cools the plume faster, which lowers the saturation vapor pressure of semi-volatile organic compounds (SVOCs) and favors partitioning to the particle phase, but also lowers the concentration of SVOCs in the plume, which favors partitioning to the gas phase (Shrivastava et al., 2006). Hence, dilution influences the predicted concentration of species (Poppe et al., 1998; Mason et al., 2001; Shrivastava et al., 2006; Donahue et al., 2006).

We use the two-parameter dilution model from Von Glasow et al. (2003) to represent plume entrainment. This model describes a rising ellipsoidal plume as it rises. The gas or aerosol species concentration dilutes according to

$$\frac{dC_{pl}}{dt} = \frac{\alpha+\beta}{t}\left(C_{bg} - C_{pl}\right), \tag{1}$$

where $C_{pl}$ and $C_{bg}$ are the species' plume and background mass concentration, respectively, $t$ is the time since emission, and $\alpha$ and $\beta$ are empirical parameters describing the dilution in the horizontal and vertical direction, respectively. The solution to this equation is

$$C_{pl}(t) = C_{bkg} - \left(C_{bkg} - C_{pl}(t_0)\right) \times \left(\frac{t_0}{t}\right)^{\alpha+\beta}, \tag{2}$$

where $t_0$ is a reference time for the emission of the plume. Because we model dilution, rather than plume shape, we set $\alpha = \beta$ since one can choose different dilution rates by varying only one of the parameters.





### 2.2.2. Specific humidity profile

Combustion of biofuel produces water due to: (a) oxidation of the hydrogen contained in the fuel, also termed 'combustion moisture', which can be calculated from the stoichiometry of the combustion reaction, and (b) the release of the water present in the fuel that is not chemically bound to the organic molecules of the fuel. The latter is also termed 'fuel moisture', expressed as a percentage of the fuel mass (Parmar et al., 2008).

Combustion moisture ranges from 0.53 to 0.83 moles of water per mole of $CO_2$ emitted, depending on the type of biofuel (Parmar et al., 2008); we use 0.72 according to the fuel composition in Nussbaumer et al. (2003). A fuel moisture (wet basis) of 50% gives a total of 1.54 kg of water emitted per kg of biofuel combusted. This corresponds to a water mixing ratio of $4 \times 10^4$ $ppm_v$ after stoichiometric combustion and complete evaporation, similar to the value Mason et al. (2001) used for a fresh biomass burning plume ($10^4$ $ppm_v$). The specific humidity profile (Figure 1a) is derived by using the dilution model (Eq. (1)) with an initial and background specific humidity of 0.15 and 0.01 kg/kg, respectively.

### 2.2.3. Temperature profile

The lower heating value of the fuel (16,000 kJ/kg; Boundy et al., 2011) combined with an energy balance gives the initial temperature of the parcel (1560 K) after stoichiometric combustion. The dilution model (Eq. (1)) provides the mass of air entrained from the background at each time step. By combining this mass balance with an energy balance, one can derive the temperature at each condition. Discretizing the plume evolution in $k = 1, 2, …, N$ time steps of width $\Delta t = 0.05$ s gives an equation for the temperature profile:

$$T_{k+1} = T_k \times \left( \frac{T_k}{T_a} \frac{\Delta t \, (\alpha + \beta)}{t_k} + \left( 1 - \frac{\Delta t \, (\alpha + \beta)}{t_k} \right) \right)^{-1}, \tag{3}$$

where $t_k$ is the time after emission at the $k$-th time step, $T_a$ is the ambient temperature, and $T_1$ is the stoichiometric combustion temperature. As explained in Sect. 2.2.1, we assume $\alpha = \beta$. Within 15 s the plume reaches ambient temperature and specific humidity (*i.e.*, plume-exit), so this time period is used for presentation.

### 2.2.4. Background species

Background aerosols and gases are brought into the plume by entrainment. The background gas composition is the same as in the work of Riemer et al. (2009), and there are no additional sources of particles or gases. Background aerosols are composed of 50% ammonium sulfate and 50% OC, are lognormally distributed with a count median diameter (CMD) of 500 nm, geometric standard deviation (GSD) of 1.3, and number concentration of $2.8 \times 10^8$ $m^{-3}$. We chose this low concentration, characteristic of a clean atmosphere, to focus on the evolution of the plume rather than the effects of background aerosols, since they can significantly affect the CCN activity of plume particles (Fierce et al., 2013).





### 2.3. Study design

About 300 PartMC-MOSAIC simulations with varying initial conditions were conducted, as summarized in Table 1. These initial conditions do not reproduce conditions exactly at the flame exit, because all reported measurements occur after some dilution and cooling. Instead, we chose initial conditions that produce plume-exit properties consistent with reported measurements. This allows an evaluation of how plume-exit properties are affected by in-plume dynamics. This section describes how initial conditions or other model parameters were varied to explore the sensitivities of plume-exit CCN. Table 1 also identifies an "illustrative case" which is used to introduce general plume behavior.

#### 2.3.1. Initial condition: particle composition

The initial composition of the plume aerosols is taken from laboratory experiments of biomass combustion of chamise, ponderosa pine, and palmetto reported by Lewis et al. (2009). Composition of biofuel emissions other than BC and OC has not been reported, and, although biomass and biofuel aerosol emissions are not identical (Hays et al., 2002), the composition is similar (Woo et al., 2003); we use a range of compositions in this work. Since potassium is not represented in PartMC-MOSAIC, all potassium was apportioned as sodium (Na) following the work of Hand et al. (2010) and Zaveri et al. (2008). Detailed composition is given in Table 2. BC/OC ratios from in-use cooking with biofuel have been measured between about 0.05 and 0.6 (Roden et al., 2006), consistent with the values used here. Aerosols from the combustion of ponderosa pine have a lower BC/OC ratio and inorganic content than those from combustion of chamise, and aerosols from combustion of palmetto have a large inorganic content. The variability in inorganic content gives a wide range of hygroscopicities, as shown in Table 2.

In the initial model input, the aerosol is assigned the composition in Table 2. Some of the constituents may partition to the gas phase at high temperatures (Knudsen et al., 2004), and MOSAIC brings the system to equilibrium within 0.2 s.

#### 2.3.2. Initial condition: particle size distribution

Particles at emission are assumed to follow a lognormal size distribution described by a Count Median Diameter (CMD) and Geometric Standard Deviation (GSD). Studies on residential stove emissions report measured CMD values between 50 and 300 nm, and GSD values between 1.3 and 2.2 (*e.g.*, Dasch et al., 1982; Hedberg et al., 2002; Tissari et al., 2007). For the initial size distribution we modeled a unimodal distribution with $CMD_0$ of 25, 50, 100, or 300 nm, and $GSD_0$ of 1.3. The corresponding initial particle number concentration was calculated using an emission factor of particulate matter ($EF_{PM}$) of 2, 4, 8, or 12 g/kg. We also modeled scenarios with $GSD_0$ of 1.1, 1.3, or 1.6, each with a $CMD_0$ of 25, 50, or 100 nm, each with $EF_{PM}$ of 4 g/kg.





### 2.3.3. Initial condition: Gases and semi-volatile organic compounds

Table 3 presents EFs of the major gas species in the plume. To assemble a plume composition from diverse reports, we used the ratio between the EF of each species and the EF of CO, and multiplied it by a central estimate of $EF_{CO}$ (42 g/kg, Akagi et al., 2011).

SVOCs affect the particle size and composition of emitted aerosols (Martin et al., 2013; Vakkari et al., 2014). The exact chemical identification of secondary organic species is not necessary for our purposes, instead only the volatility is important to describe partitioning in the plume (Donahue et al., 2006) because the short residence time in the plume precludes significant photochemical reactions. The species SVOC1 and SVOC2 in Table 3 are proxies for two SVOCs that we use to explore the effects of gases with different volatilities in the plume. MOSAIC uses the SORGAM scheme to model the formation and
partitioning of low-volatility gas species. SVOC1 and SVOC2 correspond to two of eight model species specified in SORGAM with vapor pressures of $4.0 \times 10^{-6}$ Pa and $1.2 \times 10^{-4}$ Pa at 298 K, respectively.

To model different situations of total mass released, we used two values of emission factors ($EF_{SVOC}$) for all non-methane hydrocarbons: $EF_{SVOC}$ of zero, representing a combustion event with no or negligible emissions of SVOCs, and 5.62 g/kg, a low estimate for cooking stoves (Akagi et al., 2011). Simulations with higher $EF_{SVOC}$ values lead to the same conclusions.

### 2.3.4. Plume dilution rate

Dilution rate influences the predicted concentration of species in the plume, and hence, can affect the resulting particle composition and size. We model the range of dilution that could be expected in a biofuel burning plume, represented by the parameter $\alpha$ in Eq. (1). We estimate a value of $\alpha$ between 1 and 2 by comparing modeled and measured temperature profiles of a plume from a wood heating stove, as shown in Fig. S1. This is equivalent to a dilution rate between 1.0 and 2.7 s$^{-1}$ when
the plume reaches 325 K. Values of $\alpha$ smaller than 0.5 are not likely since this would correspond to dilution rate below 0.02 s$^{-1}$ at 325 K, or a dilution ratio smaller than 5:1, and would take the plume more than 30 s to reach ambient temperature. On the other hand, values of $\alpha$ greater than 1 do not affect the mass of SVOC condensed, but reduce the coagulation rate so that changes in particle size and mixing state are less pronounced. Hence, simulations with $\alpha$ greater than 1 do not yield different CCN activity at plume-exit, although the plume reaches ambient temperature faster. Therefore, we present scenarios modeled
with $\alpha = \beta = 0.5$, 0.7, and 1.0.

### 2.3.5. Mixing state: external vs. internal mixture as initial condition

Biofuel combustion is a dynamic process, varying between flaming and smoldering. Hence, the composition of the particles emitted is not perfectly homogeneous and might change with time. To assess the effect of the initial mixing state on plume-exit CCN we used as initial condition an external mixture containing three unimodal populations: an inorganic mode composed
of ammonium sulfate, one of non-volatile OC, and one of BC. The three populations have the same $CMD_0$ and $GSD_0$, and the mass of each population follows the composition in Table 2 for $\kappa_0$ of 0.11. We compare and contrast the results of the external





mixture with a case where we assume an internal mixture of the same components as initial condition, with the same mass fractions as the external mixture case.

### 2.3.6. Role of Aitken mode particles in determining plume-exit CCN

Biofuel burning aerosols often have bimodal size distributions, containing an Aitken mode (particles less than 100 nm) and an

accumulation mode (particles larger than 100 nm and less than one micrometer) (*e.g.*, Hedberg et al., 2002; Li et al., 2007). The particles in the Aitken mode have a high coagulation rate, and inorganic species are usually more abundant in this mode (Torvela et al., 2014).

We modeled a plume in which the initial distribution contained an Aitken mode with $CMD_1 = 30$ nm and an accumulation mode with $CMD_2 = 100$ nm. We studied two different cases: an internal mixture in which particles in both modes have the

same composition, and an external mixture in which the Aitken mode is composed of only ammonium sulfate and the accumulation mode is composed of a mixture of BC and OC. We call these the 'homogeneous' and 'heterogeneous' bimodal distribution scenarios, respectively. In both cases the mass fractions of BC, OC, and ammonium sulfate of the overall population are the same. To study the effect of small particles on plume-exit CCN we also varied the initial concentration of the Aitken mode ($N_{0,Aitken}$), and changed the concentration of the accumulation mode to keep the same particulate mass emitted

($EF_{PM}$ of 4 g/kg). Since the mass of ammonium sulfate varies with $N_{0,Aitken}$, the modeled composition differs from those described in Table 2 in these cases.

We also repeated simulations with either the Aitken or accumulation mode removed, leaving only the remaining mode as a unimodal distribution. By comparing the resulting activation properties of these simulations, we assessed the role of the Aitken mode aerosols on the resulting CCN concentration.

### 2.4.  Simple estimates of condensation and coagulation

While PartMC-MOSAIC models the aerosol population in great detail, this detail may not be necessary to capture all aerosol evolution. We assess the plume processes that can be simplified or parameterized by comparing simple estimates of the evolution of particle properties with PartMC-MOSAIC predictions. These simple estimates are described below.

For coagulation, we estimated the number of particles lost at each time step using the formulation for a monodisperse aerosol

distribution (Seinfeld and Pandis, 2006),

$$\frac{dN}{dt} = 4\sqrt{\frac{6k_B TD}{\rho}}\, N^2$$

where $N$ is the particle number concentration, $k_B$ is Boltzmann's constant, T is the plume's temperature, and $D$ and $\rho$ are the particle diameter and density, respectively.



Regarding condensation, from the total mass condensed determined by PartMC-MOSAIC, we derived the consequent increase in particle size (CMD) by assuming that the GSD stays constant. We also calculated an average hygroscopicity κ for all the plume particles to assess whether it could reproduce the results obtained with PartMC-MOSAIC. With the derived CMD, average κ, and particle number concentration, we estimated the plume-exit CCN and compared with PartMC-MOSAIC predictions. This discussion is presented in Sect. 3.1.

## 3. Results

We begin the discussion of results with an illustrative case (Sect. 3.1) that demonstrates the evolution within the plume for a reference case, and then evaluate how this evolution changes with variations in the initial conditions (Sections 3.2 through 3.5). Inputs for the illustrative case are listed in Table 1.

### 3.1. Illustrative case

Figure 1b shows the evolution of the mass concentration of the main aerosol species for the illustrative case. SVOCs condense within the first two seconds of the plume's evolution, driven by the rapid drop in temperature (Figure 1a). The increase in inorganic species shortly after SVOC condensation consists mostly of $NH_4Cl$ that condenses at temperatures below 340 K (Zaveri et al., 2008). Condensation of $NO_3$ constitutes only 10% of the inorganic mass. Number concentration is also shown in Fig. 1b; it decreases faster than mass concentration due to coagulation.

The effect of the rapid condensation of SVOCs and inorganic species on particle size and hygroscopicity can be seen in Fig. 1c. CMD increases from 100 to 140 nm (black line), and the average κ of the plume aerosols increases from 0.004 to 0.06 (blue line). By repeating the simulations with either condensation or coagulation disabled and comparing the results, we conclude that about 20% of the increase in CMD can be attributed to coagulation, and the remaining 80% is from condensation of SVOCs. In this scenario, where the particles begin with a homogeneous composition, κ of individual particles changes only by condensation. Once the plume cools below 350 K (t=2.4 s), the CMD and average κ change by less than 1% and 3%, respectively. Since most particle sampling occurs at conditions below 300-325 K, aerosol properties measured at this temperature are sufficient to represent particles later in the evolution of a young plume.

Since condensation is the main cause of increased particle size and hygroscopicity, it is also the main factor driving particle activation. Coagulation increases CCN efficiency by about 20% by increasing particle size, but it also lowers plume-exit CCN by 25% due to the decrease in particle number. The increase in particle size and hygroscopicity leads to an increase in the fraction of particles that can become CCN, as shown in Fig. 1d for different $s_{sat}$ values. About half of the plume particles can activate at $s_{sat}$ of 0.3%, while most, but not all, particles activate at $s_{sat}$ of 0.5%. Regardless of the value of $s_{sat}$ chosen, $CCN(t)/N_{325}$ changes by less than 3% after the initial rapid evolution of the plume.





Transformation in this illustrative case can be summarized as a rapid increase in particle size and hygroscopicity dominated by SVOC condensation, and a decrease in particle number concentration determined by the coagulation rate and plume dilution. Since condensation is the main factor, one can estimate activation with a simple analysis as described in Sect. 2.4. Using only the condensed SVOC mass to derive the increase in particle size, and both the condensed SVOC and inorganic mass to derive

the increase in hygroscopicity, one can estimate a CCN efficiency within 15% of PartMC-MOSAIC's result, the mismatch being due to neglecting the growth by coagulation. Furthermore, using the monodisperse coagulation model to estimate the particle loss, one can estimate $CCN_{exit}/N_{325}$ within 19%. Despite the complexity of the processes occurring in a fresh plume, at least in this illustrative case they can be qualitatively described with simple models.

### 3.2. Initial size and hygroscopicity, without co-emitted SVOCs

In this section we analyze the sensitivity of plume-exit CCN to initial particle size and composition, when SVOCs are not emitted simultaneously. Figure 2a to 2c show the scenarios analyzed: $CMD_0$ of 25, 50, 100, and 300 nm, and $\kappa_0$ of 0.004, 0.11, and 0.2.

Particles in all scenarios follow an evolution similar to that of the illustrative case: a rapid increase in hygroscopicity due to condensation of inorganic species (from $\kappa_o = 0.004$, 0.11, and 0.2, to $\kappa_{325} = 0.02$, 0.13, and 0.25, respectively), and in size due

to the high coagulation rate, especially for small particles (from $CMD_0 = 25$, 50, 100, and 300 nm to $CMD_{325} = 60$, 78, 111 and 304 nm, respectively). In particular, it is not possible to obtain a $CMD_{325}$ of 50 nm or smaller with an $EF_{PM}$ above 3 g/kg due to the high coagulation rate. Below 325 K there is no more condensation, and coagulation is the only process affecting particle size. During the short time between this temperature and plume-exit, size changes by less than 5%.

Figure 2a shows the critical supersaturations of the particle populations at plume-exit for these scenarios. The mid-point is the

supersaturation at which 50% of the plume particles become CCN ($ssat_{50}$), and the error bars indicate the supersaturation at which 5% ($ssat_5$) and 95% ($ssat_{95}$) of the particles become CCN. Despite the high critical supersaturations of small particles, *e.g.*, $CMD_{325}$ below 100 nm in Fig. 2a, they can contribute substantially to CCN for supersaturations above 0.3%, as shown in Fig. 2b and Table 4. This is because more small particles are emitted for a given $EF_{PM}$. However, small particles with low inorganic content form CCN with low efficiency, with $CCN_{exit}/N_{325}$ differing by up to a factor of 7 in the scenarios with $\kappa_{325}$

of 0.02 and 0.25 (black columns in Fig. 2c).

We also conducted simulations using size distributions with $GSD_0$ of 1.1, 1.3, and 1.6, each with $CMD_0$ of 25, 50, or 100 nm. As with the CMD and $\kappa$, the GSD changes less than 5% below 325 K. Regarding CCN activity, the scenario with $GSD_0$ of 1.1 can have $EI_{CCN}$ up to twice as large as with $GSD_0$ of 1.6. This is because a wider size distribution allocates some of the mass to particles too small to become CCN.



### 3.3. Effects of co-emitted SVOCs

The scenarios described in the previous sections were also modeled with co-emitted SVOCs ($EF_{SVOC}$ of 5.62 g/kg). Results appear in Fig. 2d, 2e, and 2f. The main effect of co-emitted SVOCs is the particle size increase due to the added mass, and the consequent decrease in critical supersaturation as shown in Fig. 2d and Table 5. The change in hygroscopicity is more varied; simulations with $EF_{SVOC}$ of 5.62 g/kg and an initial hygroscopicity $\kappa_0$ of 0.004, 0.11, and 0.20 changed to $\kappa_{325}$ of 0.06, 0.11, and 0.16, respectively. Engelhart et al. (2012) observed a similar change in aged biomass burning plumes; condensation of organic species increases the hygroscopicity of hydrophobic particles, but decreases it for more hygroscopic particles. This is because SVOC condensation decreases the volume fraction of the (more hygroscopic) inorganic species. Hence, the inorganic content plays less of a role in determining the CCN activity of small particles when SVOCs are co-emitted, as can be seen by comparing Fig. 2c and 2f. Results of simulations with SVOC1 and SVOC2 are the same within a few percent.

For $EF_{SVOC}$ above about 3 g/kg, condensation dominates the in-plume increase in particle size, change in hygroscopicity, and hence, critical supersaturation. This is shown in Table 5 for $ssat_{50}$ and different values of $EF_{SVOC}$ and $EF_{PM}$. Changes in $EF_{PM}$, and hence, particle number emitted and their coagulation rate, have a small effect in $ssat_{50}$. For $EF_{SVOC}$ smaller than about 3 g/kg, particles grow more by coagulation.

These results indicate that co-emission of SVOC material can be the main factor determining the CCN activity of biofuel burning aerosols. The CCN-modifying effect may be achieved only when SVOCs and primary aerosols are emitted simultaneously; it is not enough for them to be emitted from the same fire at different times.

### 3.4. Initial mixing state

In the previous scenarios, all particles had the same composition at emission. In this section we study the importance of the initial mixing state for plume-exit CCN. As initial condition we use an external mixture of three modes, OC, BC, and ammonium sulfate, all with the same size distribution, and we analyze how results differ from the case in which the initial aerosols are internally mixed, corresponding to $\kappa_0$ of 0.11 in Table 2. Unless otherwise stated, no SVOCs are co-emitted.

The particle size distribution follows the same evolution as in the previous scenarios. The hygroscopicity, however, is much more varied. Since PartMC-MOSAIC keeps track of the mode where each particle started, as well as each coagulation event, we can derive the fraction of particles that have coagulated with a particle from another mode. This is presented in Fig. 3, which shows pairs of graphs for four scenarios, with the left figure of each pair demonstrating the evolution of coagulated particles with time, and the right figure showing the plume-exit distribution of CCN as a function of supersaturation and particle type. Fig. 3b shows that coagulated and inorganic particles contribute the most to plume-exit CCN. Particles in the OC and BC mode that do not coagulate remain too hydrophobic or too small, and contribute to CCN only at supersaturations close to 1%.





The fraction of particles that coagulate with another mode is sensitive to initial size: about 1% for $CMD_0$ of 300 nm ($CMD_{325}$ of 303 nm), 17% for $CMD_0$ of 100 nm ($CMD_{325}$ of 110 nm, Figure 3a and 3b), and 97% for $CMD_0$ of 25 nm ($CMD_{325}$ of 70 nm, Figure 3g and 3h). Slower dilution rates (Figure 3c) and higher $EF_{PM}$ (Figure 3e) also increase the coagulation between modes, but these factors affect plume-exit CCN substantially less than initial particle size, as can be seen by comparing Fig. 3d and 3f with Fig. 3h. A higher dilution rate ($\alpha = 1$) actually decreases the coagulation rate, hence it affects plume-exit CCN by just a few percent. While other research has indicated that dilution is an important factor in plume chemistry and aerosol composition (Shrivastava et al., 2006; Poppe et al., 2000; Donahue et al., 2006), their work addresses large plumes aging over hours, while our work focuses on small plumes and aging time scales of seconds.

As shown in Fig. 3a, most of the coagulation between modes occurs early in the evolution of the plume. This fraction of coagulated particles increases by less than 16% below 325 K. This implies that activation properties are set early in the plume evolution, with $ssat_{50}$ changing by only 10% between 325 K and plume exit.

Figure 4 shows how $EI_{CCN}$ differs when particles of different size are externally mixed at emission, as compared with the internal mixture case (solid and dashed lines, respectively). For $CMD_0$ of 50 nm (black lines), $EI_{CCN}$ of both mixing states is within 20%. For $CMD_0$ of 300 nm (blue lines) $EI_{CCN}$ is the same within 5% for $s_{sat}$ above 0.2%. The deviation grows with decreasing supersaturation, up to a factor of 2.5 at $s_{sat}$ of 0.1%. This low sensitivity to initial mixing state occurs because small particles have a higher coagulation rate and hence have a more homogeneous composition at plume-exit. This is consistent with Che et al. (2016), who observed that particles growing in the Aitken mode become more internally mixed in part due to coagulation. On the other hand, large particles have a low critical supersaturation, regardless of mixing state. For intermediate particles with $CMD_0$ of 100 nm (red lines) $EI_{CCN}$ differs by up to a factor of 4, indicating that particles are still externally mixed by plume-exit. However, when SVOCs are co-emitted (red dotted line), the two scenarios differ by only 5% to 20% (red dashed line), because the added SVOC mass also has a homogenizing effect on the composition of the particle population. Hence, the initial mixing state has a smaller effect on plume-exit CCN when SVOCs are co-emitted.

Figure 5 shows the distribution of critical supersaturation at plume-exit, similar to the presentation in Fig. 2. The distribution calculated using the composition from PartMC-MOSAIC is compared with the one using an average hygroscopicity for all particles. The use of an average hygroscopicity for particles that started with an external mixture leads to overestimating the $\kappa$ of OC and BC particles, and hence, their contribution to CCN. This is consistent with the field studies of Wang et al. (2010) and Che et al. (2016), who found that the internal mixture assumption overestimates CCN number by 10% to 45%. Figure 5 shows that an average hygroscopicity underestimates $ssat_{50}$ by at least 60% in the scenarios with $CMD_{325}$ of 110 nm, $\alpha$ of 0.5, or $EF_{PM}$ of 12 g/kg. Since an average $\kappa$ implies that all particles have the same hygroscopicity, its use also narrows the range of critical supersaturations (*i.e.*, $ssat_{95} - ssat_5$) by up to 80% for these scenarios. On the other hand, an average hygroscopicity is correct to within 10% for the case of small particles ($CMD_{325}$ of 70 nm, top of Fig. 5) and 29% for a case with co-emitted SVOCs ($EF_{SVOC}$ of 5.62 g/kg). Hence, there is less need to understand the initial mixing state when plume particles have either





a small size at emission, or SVOCs are co-emitted. This is consistent with the findings of Fierce et al. (2013) in their study of CCN formation from diesel emissions under atmospheric aging, that is, that CCN number is less sensitive to initial mixing state under rapid aging by condensation.

### 3.5. Role of Aitken mode particles in determining plume-exit CCN

5 In Sect. 3.2 we showed that small particles, *i.e.*, those with a $CMD_{325}$ smaller than 100 nm, can substantially contribute to plume-exit CCN (Figure 2). In this section we examine the CCN activity of these small particles when they are emitted in a bimodal distribution. We differentiate between homogeneous distributions, in which all particles have the same composition, or heterogeneous distributions, in which the Aitken mode is composed of ammonium sulfate and the accumulation mode contains BC and OC. For most of the simulations, we exclude co-emitted SVOCs in order to focus only on how the differently

10 sized particles interact. All simulations use $EF_{PM}$ of 4 g/kg.

Figure 6 shows $EI_{CCN}$ for different initial relative concentrations of the Aitken ($N_{0,Aitken}$) and accumulation mode ($N_{0,Acc}$). We use the ratio between the two ($R_{Ait} = N_{0,Aitken}/N_{0,Acc}$) to indicate it. The figure also shows whether plume-exit CCN originate in the accumulation mode, the Aitken mode, or both (colored areas). When particles are heterogeneously mixed at emission, $EI_{CCN}$ change by up to a factor of 2 from $R_{Ait}$ of 0.25 (Figure 6a) to 8.4 (Figure 6c).

15 Figure 6 also shows the CCN spectra if the same size distribution were emitted with a homogeneous composition (black dashed lines). When the homogeneous-composition curves differ from the heterogeneous scenario (area graph), then identifying compositional differences between Aitken and accumulation mode is important. Also shown in the Figure are the CCN if the accumulation mode component of the size distribution were emitted alone (red dotted lines), and if the Aitken mode component were emitted alone (blue dashed lines). Comparison between the bimodal result and the accumulation or Aitken mode spectra

20 shows how the interaction between the modes affects plume-exit CCN. When these curves differ, plume-exit CCN would differ if the two modes were emitted simultaneously or sequentially.

For $R_{Ait}$ less than 1 (Figure 6a), most of the CCN originate with accumulation-mode particles. $EI_{CCN}$ decreases by less than 17% when the simulation is repeated with the Aitken mode removed (red dashed line). On the other hand, when $N_{0,Aitken}$ is similar to $N_{0,Acc}$ (Figure 6b), most of the CCN have a component from the Aitken mode. A simulation using only the Aitken

25 mode gives $EI_{CCN}$ within a few percent of the bimodal distribution (blue dashed line), indicating that the accumulation mode does not contribute much to $EI_{CCN}$. When the Aitken-mode number concentration is much higher than that in the accumulation mode (Figure 6c), most CCN also have a component from the Aitken mode, and the presence of the accumulation mode can actually decrease $EI_{CCN}$ due to the higher coagulation rate and particle loss, as shown by the comparison with the Aitken-only simulation (blue dashed line). Increasing $N_{0,Aitken}$ increases plume-exit CCN because it increases the inorganic mass in the

30 plume (by a factor of 4 in Fig. 6c compared to Fig. 6b), and also the hygroscopicity of OC and BC particles that coagulate with inorganic particles, as discussed in Sect. 3.4.





When particles are initialized with a homogeneous composition (black lines in Fig. 6), the number of particles originally in the Aitken mode has a small effect on $EI_{CCN}$. Varying $R_{Ait}$ from the minimum (Figure 6a) to the maximum case (Figure 6c) changes $EI_{CCN}$ by 7%, because the accumulation mode is already hygroscopic enough to become CCN.

Figure 7 summarizes plume-exit properties for simulations in which the initial distribution is a bimodal, heterogeneous distribution. Figure 7a shows the critical supersaturation distribution within each particle population for the composition from PartMC-MOSAIC (blue), beginning with either a homogeneous or a heterogeneous population. The critical saturation that would be predicted with the use of an average $\kappa$ across all particles is also shown (red). Accumulation-mode (squares) and Aitken mode (triangles) are separated. In the heterogeneous case, the inorganic mass is found in the Aitken mode; artificially spreading this mass across the entire particle distribution with an average $\kappa$ leads to overestimating the critical supersaturation of the Aitken mode and underestimating that of the accumulation mode.

Figures 7b and 7c summarize $EI_{CCN}$ for two atmospherically-relevant values of $s_{sat}$: 0.3% and 0.7%. At $s_{sat}$ of 0.3% (Figure 7b), the use of an average $\kappa$ disagrees with PartMC results by 61% for $R_{Ait}$ of 8.4, to a factor of 47 for $R_{Ait}$ of 0.25. This occurs because the uncoagulated particles emitted at $R_{Ait}$ of 0.25 have a great diversity in composition. When SVOCs are co-emitted, particles become hygroscopic and also reduce the diversity in $\kappa$; the estimated $EI_{CCN}$ agrees within 48% at $s_{sat}$ of 0.3%. At higher supersaturations ($s_{sat}$ of 0.7% in Fig. 7c), most particles can function as CCN, and the disagreement in $EI_{CCN}$ is at most 27%.

## 4. Summary and conclusion

We modeled the evolution of a biofuel burning plume from emission until it reaches ambient temperature and RH, for a variety of initial conditions and plume properties, with the objective of improving estimates of emitted CCN from residential biofuel combustion. The main findings of this study are:

- After the plume dilutes and temperature drops below 350 K (t = 2.4 s for $\alpha = \beta = 0.7$), particle size and $\kappa$ values change by less than 5%. This is observed in all scenarios.
- Co-emission of SVOCs can be the main factor determining plume-exit CCN for hydrophobic or small particles, increasing $EI_{CCN}$ by up to three orders of magnitude. If particles are already large and hygroscopic enough to activate at emission, co-emission of SVOCs has a small effect on $EI_{CCN}$ (less than 5%).
- Some combinations of emission factor and size are impossible because of rapid plume coagulation. The upper limit of $EI_{CCN}$ is about $10^{16}$ $kg^{-1}$. Depending on emission factor, particle size, and composition, some of these particles may not activate at low $s_{sat}$. For $EF_{PM}$ greater than 3 g/kg, plume-exit CMD is at least 50 nm.





- When particles are emitted with smaller diameters, more CCN result from the same emitted mass, yet these particles activate at higher supersaturations. For particle diameters below 100 nm, 10-80% of the particles can serve as CCN at atmospheric supersaturations. This dependence on initial particle size and hygroscopicity is shown in Fig. 2.

- When particles are emitted as a bimodal distribution, the Aitken mode contributes less than 2% to the plume-exit CCN if both modes have the same composition.

- When the number concentration of more-hygroscopic Aitken mode particles is high compared to less-hygroscopic accumulation mode particles, the Aitken mode particles can create additional CCN through self-coagulation, but they have little effect on the CCN activity of accumulation-mode particles despite a large number of coagulation events between Aitken mode and accumulation mode particles.

- A simple model (monodisperse coagulation and average hygroscopicity) can be used to estimate plume-exit CCN within about 20% if particles are unimodal and have homogeneous composition. The simple model would underestimate particle loss by coagulation in a bimodal distribution by at least an order of magnitude.

- External mixtures become more internally mixed when particles are emitted in the Aitken mode, due to the higher coagulation rate, or with co-emitted SVOCs due to the homogenizing effect of the added mass.

- Plume-exit average hygroscopicity can be used to estimate $EI_{CCN}$ within 5-20% if particles are emitted in the Aitken mode, even if externally-mixed subpopulations exist within the Aitken mode. This is also true when SVOCs are co-emitted. The estimate improves as the particle size decreases, $EF_{PM}$ increases, or $EF_{SVOC}$ increases, since these factors enhance internal mixing in the particle population.

- On the other hand, if externally-mixed particles are emitted in the accumulation mode without SVOCs, an average $\kappa$ overestimates hygroscopicity, and hence overestimate $EI_{CCN}$ by up to a factor of 2.

This work has identified conditions under which particle populations become more homogeneous during plume processes. This homogenizing effect requires the components to be truly co-emitted. Sequential emission, even if separated by a few seconds, results in greater heterogeneity.

Atmospheric aging through coating or coagulation also affects the relationship between emitted particles and cloud influence (e.g. Fierce et al., 2014). The findings here are important because they describe (1) limitations in the emitted size distributions of primary particles, (2) the nature of directly-emitted particles near sources, before atmospheric aging has occurred, and (3) the nature of atmospheric particles when other atmospheric aging is slow.

Experimental verification of the important findings could include observations of $EI_{CCN}$ using a CCN counter, compared with gaseous carbon emissions, to demonstrate that $EI_{CCN}$ is limited to $10^{16}$ kg$^{-1}$. The effect of co-emitted SVOCs could be evaluated by comparing CCN activity of a plume at different dilution ratios, where SVOCs would be expected to partition to the gas phase in the most dilute environments. The conditions under which the average-hygroscopicity assumption is expected to fail could be confirmed by performing CCN closure studies on aerosol generated with those specific characteristics; that is, aerosol



size and composition could be measured with an Aerodyne Mass Spectrometer, and then CCN predicted from those values could be compared to measurements. Because biofuel emissions vary during the course of combustion, and because simultaneous emission affects the particle properties, these experiments should be conducted in real-time or on emissions from isolated phases of burning.

## 5. Acknowledgements

F. Mena would like to thank the Fulbright Doctoral Program for their support in this research, Dr. Laura Fierce for her insightful comments, and Brent Biernbaum, Dylan Hayden, Eric McCoy, and Bill Riewerts for their experimental data of a plume temperature profile. This work was supported by the Department of Energy under grant DE-SC0006689.

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





**Table 1. Range of initial conditions and plume parameters analyzed to determine sensitivity. The "Illustrative Case" is a simulation used to present the general features observed in the plume evolution.**

| Parameter | Scenarios | Illustrative Case |
|---|---|---|
| Initial hygroscopicity | $\kappa_0 = 0.004$ (ponderosa pine), $\kappa_0 = 0.11$ (chamise), $\kappa_0 = 0.20$ (palmetto) | $\kappa_0 = 0.004$ |
| Initial CMD and GSD | $CMD_0 = 25, 50, 100, 300$ nm<br>$GSD_0 = 1.1, 1.3, 1.6$ | 100 nm<br>1.3 |
| $EF_{PM}$ | 2, 4, 8, 12 g/kg | 4 g/kg |
| Dilution rate | $\alpha = \beta = 0.5, 0.7, 1.0$ | 0.7 |
| SVOC volatility | SVOC1 (saturation vapor pressure[1] $= 4.0 \times 10^{-6}$ Pa) or SVOC2 (saturation vapor pressure[1] $= 1.2 \times 10^{-4}$ Pa) | SVOC1 |
| SVOC emission factor ($EF_{SVOC}$) | 0 (no SVOCs), 5.62 g/kg | 5.62 g/kg |
| Initial aerosol distribution | Unimodal or bimodal distribution | Unimodal |
| Initial mixing state | External mixture of OC, BC, and ammonium sulfate, or internally mixed | Internal mixture |

[1]saturation vapor pressure at 298 K





**Table 2. Initial composition of aerosols emitted from the combustion of three fuels, from Lewis et al. (2009), and the corresponding volume-average hygroscopicity parameter κ (Petters and Kreidenweis, 2007). OC specified here is assumed to be non-volatile.**

| Component | Mass Fraction [%] | | |
|---|---|---|---|
| | Ponderosa Pine | Chamise | Palmetto |
| OC | 97.8 | 48.8 | 42.8 |
| BC | 1.2 | 27 | 4.4 |
| $SO_4$ | 0.25 | 12.1 | 1.32 |
| $NO_3$ | 0.25 | 1.21 | 0 |
| Cl | 0.25 | 4.84 | 33 |
| $NH_4$ | 0.12 | 0 | 13.2 |
| K | 0.1 | 6.05 | 2.64 |
| Na | 0.03 | 0 | 2.64 |
| $\kappa_0$ | 0.004 | 0.11 | 0.20 |

**Table 3. Emission factor of plume gases**

| Gas | Emission Factor [g/kg] | Reference |
|---|---|---|
| $CO_2$ | 1626 | Akagi et al. (2011) |
| $NO_x$ (as $NO_2$) | 2.04 | Christian et al. (2010) |
| CO | 42 | Akagi et al. (2011) |
| $NH_3$ | 0.947 | Akagi et al. (2011) |
| $SO_2$ | 0.57 | Burling et al. (2010) |
| HCl | 0.139 | Burling et al. (2010) |
| $CH_4$ | 2.32 | Akagi et al. (2011) |
| SVOC1, SVOC2 | 0, 5.62 | Akagi et al., (2011) |





**Table 4. $EI_{CCN}$ [kg$^{-1}$] at $s_{sat}$ of 0.7% for scenarios with $CMD_0$ = 25, 50, 100, and 300 nm (*i.e.*, $CMD_{325}$ = 60, 78, 111, and 304 nm), and $\kappa_0$ = 0.004, 0.11, and 0.2 ($\kappa_{325}$ = 0.02, 0.13, and 0.25), and no co-emitted SVOCs.**

| $EI_{CCN}$ at a $s_{sat}$ of 0.7% ($\times 10^{15}$) | | | |
|---|---|---|---|
| | $\kappa_0$ | | |
| $CMD_0$ [nm] | 0.004 | 0.11 | 0.2 |
| 25 | 1.05 | 5.73 | 7.44 |
| 50 | 1.86 | 6.54 | 7.84 |
| 100 | 2.57 | 3.46 | 3.14 |
| 300 | 0.21 | 0.17 | 0.15 |

5 **Table 5. Median critical supersaturation ($ssat_{50}$) from simulations with $EF_{PM}$ of 2, 4, or 8 g/kg, and $EF_{SVOC}$ of 0, 3, 5.62, and 10 g/kg. An initial size of $CMD_0$ of 25 nm was used to model a high coagulation rate.**

| $ssat_{50}$ [%] | | | | |
|---|---|---|---|---|
| | $EF_{SVOC}$ [g/kg] | | | |
| $EF_{PM}$ [g/kg] | 0 | 3 | 5.62 | 10 |
| 2 | >1 | 0.65 | 0.48 | 0.36 |
| 4 | >1 | 0.64 | 0.48 | 0.38 |
| 8 | 0.91 | 0.63 | 0.49 | 0.38 |




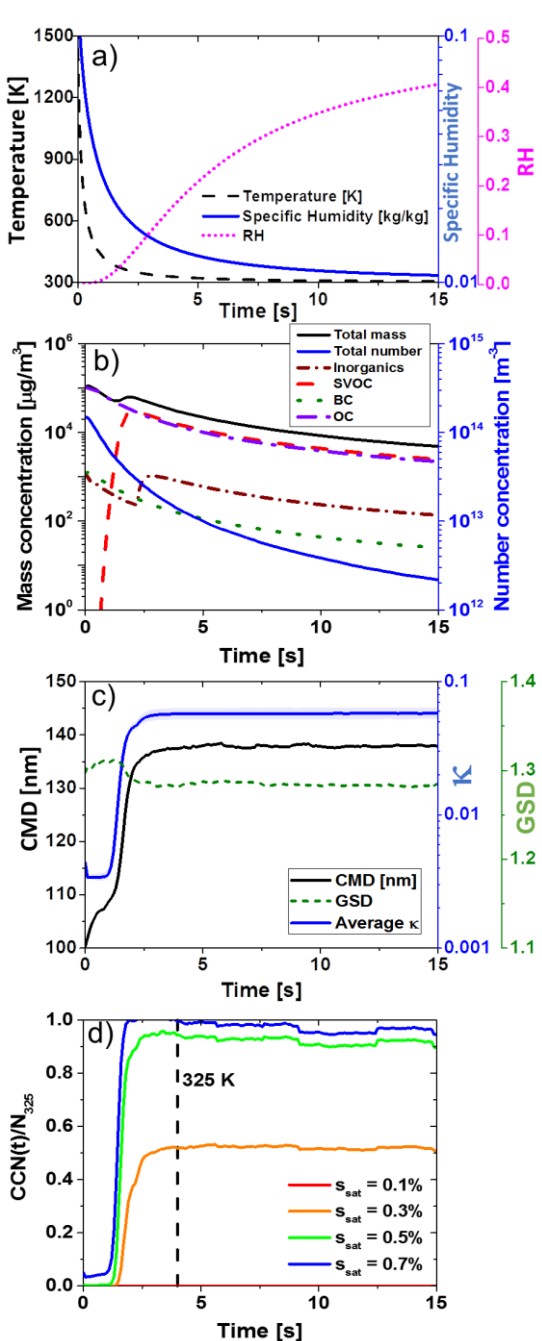

**Figure 1. (a) Prescribed profiles of specific humidity (left axis) and temperature (right axis). From these profiles, PartMC-MOSAIC calculates the plume RH (right axis). Time series of (b) mass concentration of plume aerosol species (left axis) and number concentration (right axis), (c) CMD (left axis), GSD and the average $\kappa$ of all plume particles (right axis). The shaded area in $\kappa$ is one standard deviation from the average. (d) $CCN(t)/N_{325}$ for $s_{sat}$ of 0.1, 0.3, 0.5, and 0.7%. The vertical black line highlights when the plume temperature is 325 K. Although we modeled the first 30 seconds, we present results until the plume reaches ambient temperature and RH (plume-exit), *i.e.*, t = 15 s.**





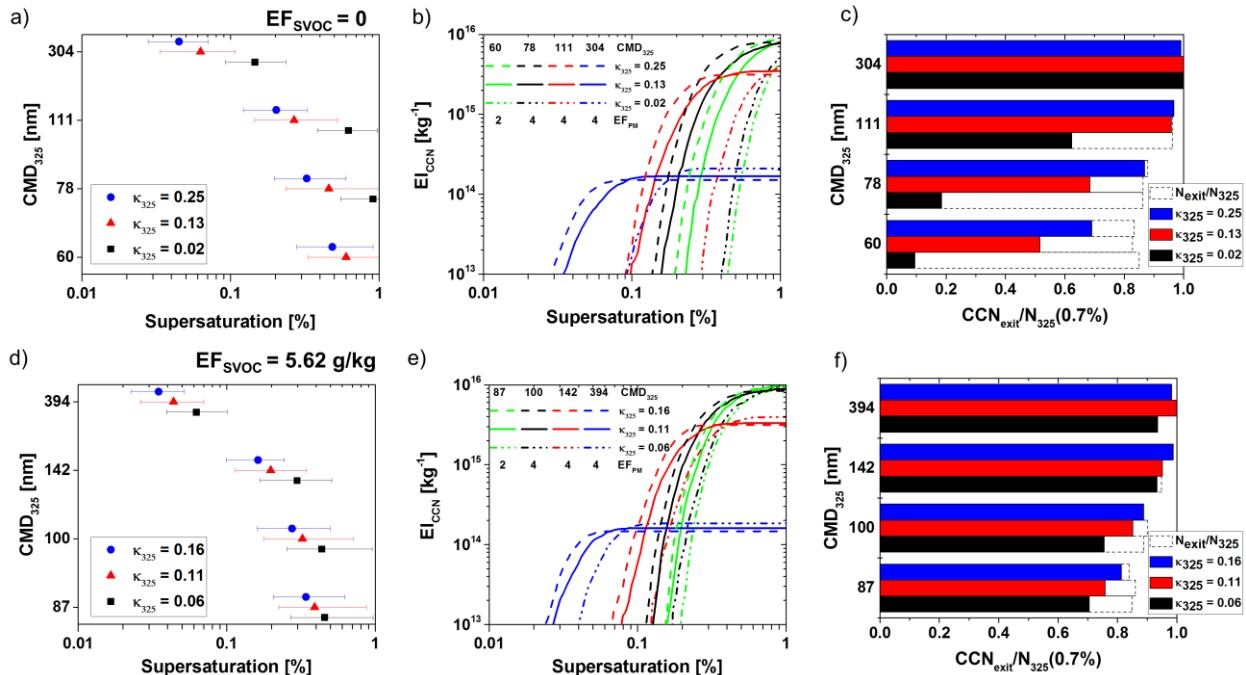

**Figure 2. (a)** Critical supersaturation of particle populations ($ssat_5$, $ssat_{50}$, and $ssat_{95}$), **(b)** $EI_{CCN}$, and **(c)** $CCN_{exit}/N_{325}$ at $s_{sat}$ of 0.7% for four different initial sizes ($CMD_0$ = 25, 50, 100, and 300 nm) and compositions ($\kappa_0$ = 0.004, 0.11, and 0.20). Labels show CMD and $\kappa$ at 325 K. No SVOCs are co-emitted ($EF_{SVOC}$ = 0). Figures **(d)**, **(e)**, and **(f)** show the same scenarios but for $EF_{SVOC}$ = 5.62 g/kg. The scenario with $CMD_0$ = 25 nm uses $EF_{PM}$ of 2 instead of 4 g/kg to achieve smaller $CMD_{325}$ at plume exit.

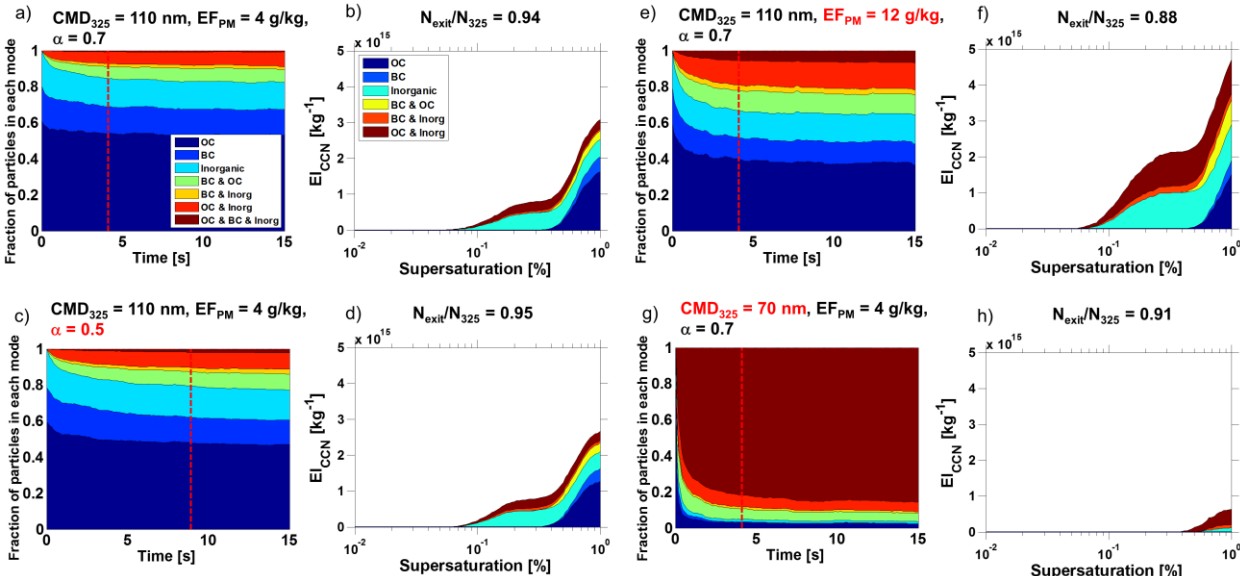

**Figure 3.** Fraction of particles in each mode (OC, BC, inorganic, or coagulated) as the plume evolves, showing how different initial conditions affect the coagulation between modes in an external mixture. **a)** Scenario with $CMD_0$ of 100 nm and $GSD_0$ of 1.3, $EF_{PM}$ of 4 g/kg, $\alpha$ of 0.7, and no SVOCs. The other figures are the same scenario modified with **(c)** ~~a~~ slower dilution ($\alpha$ = 0.5), **(e)** $EF_{PM}$ of 12 g/kg, or **(g)** $CMD_0$= 25 nm ($CMD_{325}$= 70 nm). The vertical red line highlights when the plume temperature is 325 K. Figures **(b)**, **(d)**, **(f)**, and **(h)** are the corresponding $EI_{CCN}$ graphs to scenarios **(a)**, **(c)**, **(e)**, and **(g)**, respectively.





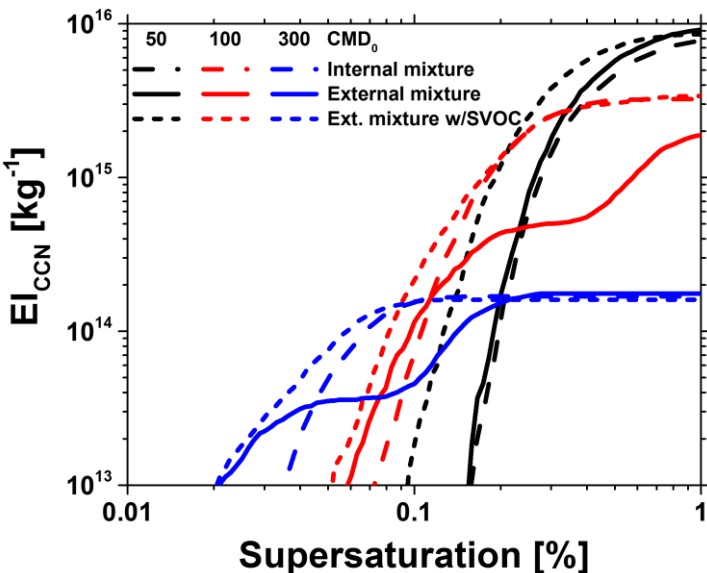

**Figure 4.** $EI_{CCN}$ for different initial mixing states and initial $CMD_0$ of 50 nm (black), 100 nm (red), or 300 nm (blue). In the internal mixture case (dashed lines) all particles have the same initial composition, and no co-emitted SVOCs. The external mixture scenarios (solid lines) have an initial composition of three modes (OC, BC, and inorganic) with the same mass fraction as in the internal mixture, and no co-emitted SVOCs. The dotted line is the same external mixture scenario but now with co-emitted SVOCs ($EF_{SVOC}$ = 5.62 g/kg).

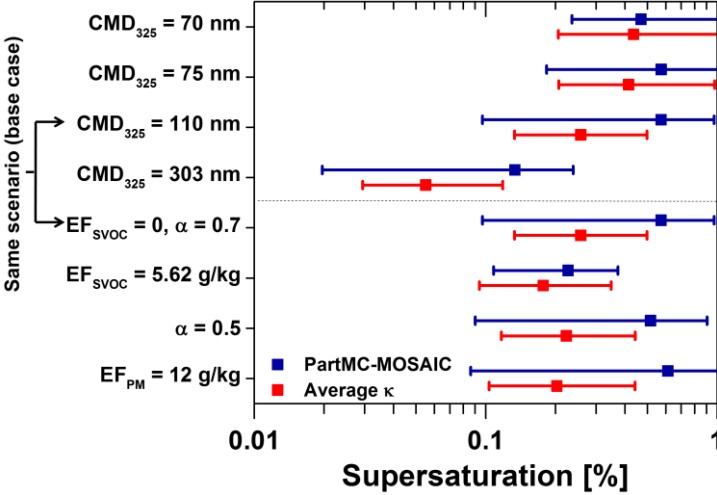

**Figure 5.** Critical supersaturation of particle populations ($ssat_5$, $ssat_{50}$, and $ssat_{95}$), calculated with either the particle composition from PartMC-MOSAIC (blue), or an average $\kappa$ for all plume particles (red). The arrow highlights a "base case" simulation, *i.e.*, an external mixture with $EP_{PM}$ of 4 g/kg, $CMD_{325}$ of 110 nm, $\alpha$ = 0.7, and $EF_{SVOC}$ = 0. All other cases shown here are variations of this base case, obtained by changing one of the parameters as indicated in the labels. The horizontal dotted line separates scenarios testing the sensitivity to initial size (above), or SVOCs, dilution rate, or $EF_{PM}$ (below).



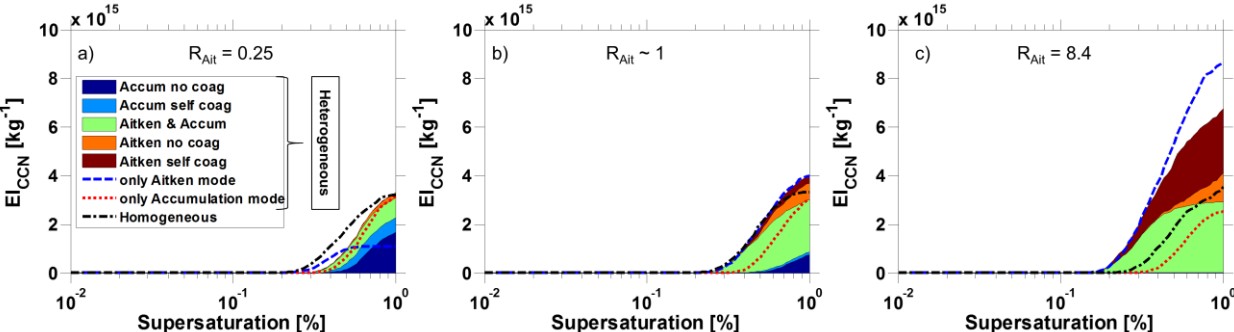

**Figure 6. EI$_{CCN}$ for three different R$_{Ait}$ = N$_{0.Ait}$/N$_{0,Acc}$ values. The scenario with R$_{Ait}$ ~ 1 has N$_{0,Aitken}$ = 1.6x10$^{14}$ m$^{-3}$ and N$_{0,Acc}$ = 1.2x10$^{14}$**
**m$^{-3}$. The colored area corresponds to a heterogeneous bimodal distribution, each color describing the source of the particle that is**
**CCN-active at plume exit. The label 'self coag' refers to particles that have coagulated only with particles from the same mode. 'no**
5 **coag' refers to particles that have not coagulated with any other particle type. The dashed blue and red lines correspond to a**
**repetition of the simulation but with either the accumulation or Aitken mode removed, respectively. The black line corresponds to**
**a homogeneous bimodal distribution, using the same R$_{Ait}$ values.**

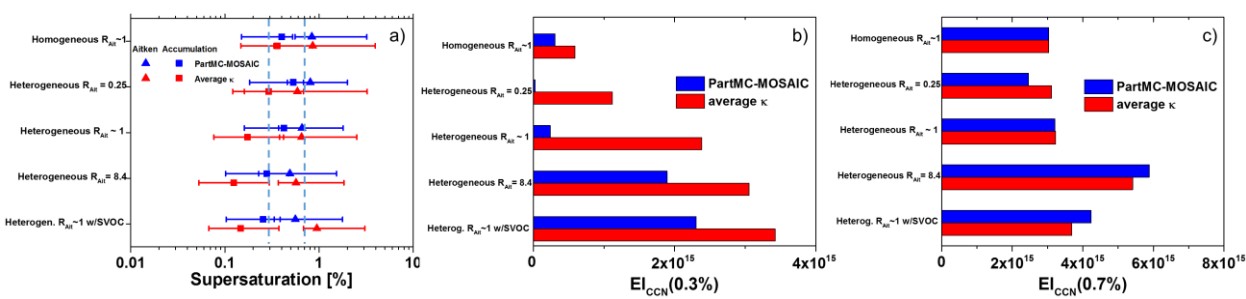

**Figure 7. a) ssat$_5$, ssat$_{50}$, and ssat$_{95}$ of homogeneous and heterogeneous bimodal distributions with different R$_{Ait}$ values,**
**calculated separately for the Aitken mode (triangles) and accumulation mode (squares). Calculations were made with**
**either the composition from PartMC-MOSAIC (blue), or using an average κ for all particles (red). The vertical blue**
**lines highlight s$_{sat}$ of 0.3% and 0.7%. b) EI$_{CCN}$ for the same scenarios for s$_{sat}$ of 0.3%, and (c) s$_{sat}$ of 0.7%.**