# Peer review of "Plume-exit modeling to determine cloud condensation nuclei activity of aerosols from residential biofuel combustion"

_Atmospheric Chemistry and Physics, 2017_

## Author Comment (AC1) · 15 Mar 2017

Figure 3b, 3d, 3f, and 3h did not include one set of coagulated particles that have activated (labeled "OC & BC & Inorg" in Fig. 3a). The attached figure corrects this, but only Fig. 3h changes notably. This change does not affect the analysis or reported findings, since it is consistent with the discussion in the manuscript. The caption of the Figure does not change.

[Figure]

[Figure]

**Fig. 1.**

---

## Referee Comment (RC1) · Anonymous Referee #1 · 21 Mar 2017

This paper describes a study of the plume exit from residential biofuel combustion, used to provide energy. The authors investigate how CCN activity varies with different assumptions about parameters, such as particle sizes, compositions, size distribution properties (mono or bimodal distributions), co-emission, mixing states and emission factors. This paper is of high scientific quality and very well written. Variations in plume exits can have various impacts on cloud formation and on aerosol indirect effects, and the findings in this paper improve the understanding of plume exits under certain conditions.

I think this paper should be published and I have only minor comments.

Page 1, line 25: I suggest adding "for energy production" (or something similar) after

the parenthesis. Page2, line 5: I suggest adding a reference to Twomey 1974. Page 5, line 5: What is the reason for choosing 325K as the reference temperature? Page 7, line 17. I suggest adding the individual $\kappa$ values used to calculate the average hygroscopicity in Table 2. Page 8, line 10. Add "the" so that "....correspond to two of the eight model species...." Page 8, line31. What about using Chamise instead of $\kappa0$? Page 11, line 11. I suggest that there were 12 scenarios analyzed ( combination of the four CMD and three $\kappa$ values) Page 11, line 14: Is $\kappa325$ the same for all CMD0? Page 11, line 22: I suggest adding: "...they can contribute substantially to the EI of CCN Page 11, lines 22-25. I believe this sentence is incomplete. Page 14, line 12. I suggest replacing "to indicate it" with "to indicate this"
* * *

---

## Referee Comment (RC2) · J. Reid (Referee) · 2 Apr 2017

This paper presents a pretty straightforward set of "Toy model" simulations of near emission aerosol thermodynamics spun towards the question of what happens to bio-fuel particles in the seconds after it leave the stack and what this potentially means for CCN activity. They provide a series of conclusions that really boil down to that condensation of semi-volatiles trumps coagulation in changing particle size which then leads to increased CCN efficiency. They use a fairly set of good modeling tools, but in a box model format where dilution and SVOC properties are explicit. They also use some extreme values of inorganic mass fractions taken from the FLAME studies to see how varying core particle hygroscopicity may change things. They also spend some time

describing results on Aiken mode particles which can be high in number but due to their size generally have low CCN efficiency unless they are highly hygroscopic.

I think for what the paper is, it is well done and generally well written. I do however have some questions of representativeness that I think can be easily fixed in the introduction and conclusions. First, the conclusions are really not that surprising and can be inferred from other studies over the past 30 years. Vanderlei Martins and I both did back of the envelope calculations >25 years ago in our papers and dissertations noting that condensation was the big dog in evolving plumes. The current paper does a nice job formalizing this in particular for near field biofuel emissions, but even here simultaneous coagulations and condensation has been studied for a very long time. I went to web of science and typed various combinations of such key words as aerosol dynamics, combustion, flue, condensation and coagulation and found a number of useful papers. This is all done for the power industry, which I guess biofuel is. Not exactly exhaust fuel for biofuel and the implications for CCN, but the key aerosol dynamics are in there and even some lab-model intercomparisons. Thus, I think a little perspective warranted on how this work bridges these fundamental thermodynamic studies with emissions to the atmosphere. The paper is somewhat unique on the CCN angle, but again some perspective is required. Even though particle microphysics and chemistry largely freezes after 5 seconds, the CCN properties of the particles will likely evolve further in the atmosphere when they mx and react with everything else. This has been a common criticism of mine of studies like the FLAME series. I think they are great for insight, but cannot be taken and directly applied to ambient aerosol populations. So what do the authors think the bottom line ramifications of their results are? Seems like we better sort out what is going on in the SVOC. It might help if this paper is torqued to frame such conclusions as priorities for studies.

Even though the thermodynamics presented here is for the immediate emissions of biofuels it might be worth joining this with the bigger picture. For example, even though these are small plumes, it might be worth the author's time to quickly go through Kip

Carrico's recent paper on near field plume behavior. I think there is some overlap in results, particularly on the laboratory side. Carrico, C. M., A. J. Prenni, S. M. Kreidenweis, E. J. T. Levin, C. S. McCluskey, P. J. DeMott, G. R. McMeeking, S. Nakao, C. Stockwell, and R. J. Yokelson (2016), Rapidly evolving ultrafine and fine mode biomass smoke physical properties: Comparing laboratory and field results, J. Geophys. Res. Atmos., 121, 5750–5768, doi:10.1002/2015JD024389. Or even go back to Vanderlei and my original work. Further back is all of the good work done by Turko.

One other minor thing, Page 7, line 10: Table 1 says "organic carbon" but I think you want to say particulate organic matter. Thus throughout the paper, there is some confusion as to what is OC and what is POM. This makes a pretty big difference in interpretation.

Hope this helps, Jeffrey S. Reid US Naval Research Laboratory

---

## Author Comment (AC2) · 31 May 2017

**Responses to Reviewer #1**

The authors thank the referee for the detailed and helpful comments. The referee comments are in italics, and our responses are in normal font.

*1. This paper describes a study of the plume exit from residential biofuel combustion, used to provide energy. The authors investigate how CCN activity varies with different assumptions about parameters, such as particle sizes, compositions, size distribution properties (mono or bimodal distributions), co-emission, mixing states and emission factors. This paper is of high scientific quality and very well written. Variations in plume exits can have various impacts on cloud formation and on aerosol indirect effects, and the findings in this paper improve the understanding of plume exits under certain conditions. I think this paper should be published and I have only minor comments.*

Thank you for summarizing the paper.

*2. Page 1, line 25: I suggest adding "for energy production" (or something similar) after the parenthesis.*

We have updated the text as suggested

*3. Page2, line 5: I suggest adding a reference to Twomey 1974.*

We have added the reference in the text

*4. Page 5, line 16: What is the reason for choosing 325 K as the reference temperature?*

This temperature was used as representative of sampling conditions, although any value close to ambient conditions would be sufficient. For clarity, we added the sentence "although any value close to ambient conditions would suffice" to Page 5, line 16.

*5. Page 7, line 25. I suggest adding the individual $\kappa$ values used to calculate the average hygroscopicity in Table 2*

We have updated Table 2 as suggested.

*6. Page 8, line 19. Add "the" so that "…correspond to two of the eight model species…"*

We have updated the text as suggested

*7. Page 9, line 9. What about using Chamise instead of $\kappa_0$?*

We have updated the text to say: "…the composition of Chamise in Table 2 ($\kappa_0$ of 0.11)."

*8. Page 11, line 22. I suggest that there were 12 scenarios analyzed (combination of the four CMD and three κ values)*

We have updated the text to say: "Figure 2a to 2c show the 12 scenarios analyzed: combinations of $CMD_0$ of 25, 50, 100, and 300 nm, with $\kappa_0$ of 0.004, 0.11, and 0.2."

*9. Page 11, line 25: Is $\kappa_{325}$ the same for all $CMD_0$?*

$\kappa_{325}$ is the same within 0.01 for all $CMD_0$. In line 25 we have added "with $\kappa_{325}$ the same within 0.01 for the different $CMD_0$", and in line 27 we have added "with $CMD_{325}$ the same within 5 nm for the different $\kappa_0$"

*10. Page 12, line 4: I suggest adding: "…they can contribute substantially to the EI of CCN"*

We have updated the text accordingly.

*11. Page 12, lines 4-7. I believe this sentence is incomplete.*

We have added "…, which eventually can become CCN."

*12. Page 14, line 26. I suggest replacing "to indicate it" with "to indicate this".*

We have updated the text accordingly.

---

## Author Comment (AC4) · 31 May 2017

**Responses to Reviewer #2**

The authors thank the referee for the detailed and helpful comments. The referee comments are in italics, and our responses are in normal font.

*1. This paper presents a pretty straightforward set of "Toy model" simulations of near emission aerosol thermodynamics spun towards the question of what happens to biofuel particles in the seconds after it leave the stack and what this potentially means for CCN activity. They provide a series of conclusions that really boil down to that condensation of semi-volatiles trumps coagulation in changing particle size which then leads to increased CCN efficiency. They use a fairly set of good modeling tools, but in a box model format where dilution and SVOC properties are explicit. They also use some extreme values of inorganic mass fractions taken from the FLAME studies to see how varying core particle hygroscopicity may change things. They also spend some time describing results on Aiken mode particles which can be high in number but due to their size generally have low CCN efficiency unless they are highly hygroscopic.*

Thank you for summarizing the paper.

*2. I think for what the paper is, it is well done and generally well written. I do however have some questions of representativeness that I think can be easily fixed in the introduction and conclusions. First, the conclusions are really not that surprising and can be inferred from other studies over the past 30 years. Vanderlei Martins and I both did back of the envelope calculations >25 years ago in our papers and dissertations noting that condensation was the big dog in evolving plumes.*

We agree that the important role of the condensation of semi-volatile species on CCN activation is not a new finding. Our contribution is on providing quantification of its role in the evolution of the mixing state of the particle population and their CCN activity.

*3. The current paper does a nice job formalizing this in particular for near field biofuel emissions, but even here simultaneous coagulations and condensation has been studied for a very long time. I went to web of science and typed various combinations of such key words as aerosol dynamics, combustion, flue, condensation and coagulation and found a number of useful papers.*

Indeed, there is plenty of research on modeling the aerosol dynamics in young plumes. In page 3, lines 5 to 15 we cite the work of Alvarado, Hobbs, Gao, Mason, and Trentmann for the case of biomass burning. The important distinction with plumes from residential biofuel combustion is that the small scale and rapid dilution to ambient conditions can affect the aerosol dynamics, since dilution controls the concentration of species in the plume, and hence, coagulation and condensation. Thus our modeling approach differs from previous work by modeling these plume-scale processes for the specific case of these smaller plumes, and their effect on CCN activity.

To make this point clearer in the Introduction, in page 3 line 16 we have added: "Furthermore, a distinction of our modeling approach from previous research is that we use a particle-resolved model to quantify the effect of plumescale processes on the evolution of mixing state, which can contribute to the uncertainty in estimated CCN (Ching et al., 2012)."

*4. This is all done for the power industry, which I guess biofuel is. Not exactly exhaust fuel for biofuel and the implications for CCN, but the key aerosol dynamics are in there and even some lab-model intercomparisons.*

The reviewer appears to have a misunderstanding about the source we examine as compared with previous literature. There has indeed been a lot of work describing power-plant plumes, which are large point sources emitted at relatively high altitudes. The situation we describe is for ground-level plumes with smaller extent, possibly different dilution rates, and much higher fraction of organic matter, which can condense and evaporate. These differences are the reason we rely on new simulations rather than previous work.

A second distinction between our modeling approach and previous research is a detailed description of the evolution of the particle mixing state. Other work using moment-based or sectional schemes to represent the particle population simplifies the representation of the mixing state, which contributes to the uncertainty in estimated CCN emissions. Ching et al. (2012) showed that simplifications in the representation of the particle mixing state can results in errors of up to 34% in the estimated fraction of particles that become cloud droplets, in particular for fresh emissions.

We agree that the broad conclusions could be inferred from other work, and are perhaps not surprising in retrospect. However, in papers that report CCN activity of combustion emissions, we do not generally find discussions of co-emitted semi-volatile material as important in determining CCN activity. The purpose of this paper is to provide an overview and quantification of the affecting factors so that they can be considered in future measurements and modeling. This systematic treatment differs from anecdotal inferences.

*5. Thus, I think a little perspective warranted on how this work bridges these fundamental thermodynamic studies with emissions to the atmosphere. The paper is somewhat unique on the CCN angle, but again some perspective is required. Even though particle microphysics and chemistry largely freezes after 5 seconds, the CCN properties of the particles will likely evolve further in the atmosphere when they mix and react with everything else.*

Our research addresses the initial evolution of the plume where the plume-scale processes determine the particle size and mixing state, and hence, CCN activity. These findings are meant to provide the end-point of the plume and the beginning of atmospheric evolution where, as the referee mentions, the particle properties are determined by not only within-plume processes, but also due factors external to the plume, *e.g.*, mixing with other sources or atmospheric oxidation.

In page 17, line 17, we have added: "The results given here characterize the nature of the particles at plume-exit, after their evolution in a high-concentration environment where interactions with co-emissions prevail. The work of Fierce et al. (2016) describes subsequent evolution in the atmosphere, where factors external to the plume also play

a role. Taken together, this body of work describes the complete evolution of biofuel-burning emissions beginning at the source and continuing through the atmosphere."

*6. This has been a common criticism of mine of studies like the FLAME series. I think they are great for insight, but cannot be taken and directly applied to ambient aerosol populations. So what do the authors think the bottom line ramifications of their results are?*

We have now added a summary of the major findings at the end of the bullet-point list in the "Summary and Conclusion" section. In page 17, line 3, we have added: "In summary, this work has systematically evaluated how and when plume processes alter particle populations, thus identifying two key cautions about using measured particle properties to represent CCN activity in atmospheric models. The first condition involves SVOCs, which homogenize particle populations and increase their CCN activity. This homogenization occurs only when SVOCs are truly co-emitted. Sequential emission, even if separated by a few seconds, results in greater heterogeneity; emissions from an entire burn cycle captured in an aging chamber may not represent plume-exit properties. The second caution occurs when there is a substantial Aitken mode. The Aitken members of a bimodal distribution likely never contribute to CCN activity, and when an Aitken-only mode is emitted, the plume-exit numbers are reduced by coagulation."

*7. Seems like we better sort out what is going on in the SVOC. It might help if this paper is torqued to frame such conclusions as priorities for studies.*

In the summary we reference above in response to Comment 6, we have now emphasized that two processes (condensation of SVOCs and behavior of Aitken-mode particles) are important when using measured particle properties to represent particles in models. We hope this addition communicates that it would be useful to account for these processes in future measurement studies. We did not change the abstract because we believe that these findings are already emphasized sufficiently there.

*8. Even though the thermodynamics presented here is for the immediate emissions of biofuels it might be worth joining this with the bigger picture. For example, even though these are small plumes, it might be worth the author's time to quickly go through Carrico et al. (2016) recent paper on near field plume behavior. I think there is some overlap in results, particularly on the laboratory side. Carrico, C. M., A. J. Prenni, S. M. Kreidenweis, E. J. T. Levin, C. S. McCluskey, P. J. DeMott, G. R. McMeeking, S. Nakao, C. Stockwell, and R. J. Yokelson (2016), Rapidly evolving ultrafine and fine mode biomass smoke physical properties: Comparing laboratory and field results, J. Geophys. Res. Atmos., 121, 5750–5768, doi:10.1002/2015JD024389. Or even go back to Vanderlei and my original work. Further back is all of the good work done by Turko.*

Thank you for the interesting paper recommendations. Carrico et al. (2016) measured changes in particle size and optical properties with combustion phase (flaming vs. smoldering) from the combustion of biomass, in both fresh and aged plumes. Carrico et al. (2016) make an interesting connection between the measured particle properties at

emission with to those in more aged plumes. One interesting overlap is that they report a particle size growth due to coagulation from 50 nm to 100 nm after 3 hours of aging, which is similar to the growth predicted in section 3.2. However, since they do not measure or model the evolution of the particle size distribution as the plume ages, and due to the larger scale of their plumes, a direct comparison with our study is not straightforward, and without a more clear connection, a comparison with their results might lead to confusion.

To connect our findings with previous research, in page 17 line 22 we have added: "Previous research has described how aerosol dynamics and atmospheric aging in young biomass burning plumes affect particle properties in the time scale of hours or days (*e.g.*, Reid et al., 1999; Trentmann et al., 2002; Alvarado and Prinn, 2009). We have shown that for residential biofuel burning plumes the plume-scale processes affect the CCN activity within seconds after emission due to the high concentration of particles and gases, and the rapid dilution to ambient conditions. We also found that plume-exit CCN concentrations change by a few percent for the range of dilutions modeled here, whereas previous research on large, aged biomass burning plumes highlights the importance of dilution rate on plume chemistry and aerosol composition (Poppe et al., 2000; Shrivastava et al., 2006; Donahue et al., 2006). Furthermore, by using a particle-resolved model, we have quantified the homogenizing effect of condensation and coagulation in these small plumes."

And to be clearer in the Introduction regarding previous research, we have adjusted the text page 3 line 9 to say: "Previous research has shown that biomass burning particles can be effective CCN. Warner and Twomey (1967), Hobbs and Radke (1969), and Holle (1971) measured increases in CCN concentrations due to biomass burning emissions. Field studies have measured the particle evolution in biomass burning plumes from minutes to hours or days old (*e.g.*, Liousse et al., 1995; Reid and Hobbs, 1998; Reid et al., 1998; Reid et al., 1999), and modeling efforts have quantified the particle evolution due to coagulation, gas-particle partition, and chemical reactions (*e.g.*, Mason et al., 2001; Trentmann et al., 2002, 2005, 2006; Gao et al., 2003; Jost et al., 2003; Alvarado and Prinn, 2009)."

*9. One other minor thing, Page 7, line 10: Table 1 says "organic carbon" but I think you want to say particulate organic matter. Thus throughout the paper, there is some confusion as to what is OC and what is POM. This makes a pretty big difference in interpretation.*

Indeed, Lewis et al. (2009) reported organic mass instead of organic carbon, and we modeled POM. To avoid confusion, we have changed instances of organic carbon (OC) to primary organic aerosol (POA) in the paper, as it is referred in PartMC-MOSAIC (Riemer et al., 2009). The changes from OC to POA are in page 7 line 7, page 7 line 21 and 25, page 9 line 8, 20, and 21, page 13 line 3 and 12, page 14 line 8 and 23, page 15 line 14, Table 1, Table 2, Fig. 3, and caption of Fig. 4. We have also added in page 7 line 25: "Lewis et al. (2009) reported organic mass, which we use as POA for the initial composition in our simulations (Table 2)".

[revised manuscript text omitted]

---

## Author Comment (AC5) · 31 May 2017

The marked-up version of the manuscript with changes according to the referee's comments is attached as a supplement.

Please also note the supplement to this comment:
http://www.atmos-chem-phys-discuss.net/acp-2017-67/acp-2017-67-AC5-supplement.pdf